# A Systematic Review of Responses, Attitudes, and Utilization Behaviors on Generative AI for Teaching and Learning in Higher Education

**DOI:** 10.3390/bs15040467

**Published:** 2025-04-04

**Authors:** Fan Wu, Yang Dang, Manli Li

**Affiliations:** 1Institute of Higher Education, Shanghai University, Shanghai 200444, China; wfwufan@shu.edu.cn; 2Institute of Education, Tsinghua University, Beijing 100084, China; dangy23@mails.tsinghua.edu.cn

**Keywords:** generative AI, responses, attitudes, behaviors, higher education classroom

## Abstract

The utilization of Generative AI (GenAI) in higher education classrooms has significantly increased in recent years. Studies show that GenAI holds promise in impacting the learning experiences of both students and teachers, offering personalized learning and assessment opportunities. This study conducts a systematic review of the responses, attitudes, and behaviors related to the application of GenAI within higher education classrooms. To this end, we synthesized 99 papers published between 2020 and August 2024, focusing on the utilization of GenAI in higher education settings. The analysis addresses three key inquiries: responses, attitudes, and behaviors. This systematic review provides an updated understanding from psychological perspectives of GenAI’s role in the teaching and learning processes of higher education, with a particular emphasis on GenAI technologies.

## 1. Introduction

Generative artificial intelligence (GenAI) is not a new term but entered the public consciousness in the 2020s ([106]). In 1964, MIT computer scientists Joseph Weizenbaum developed ELIZA, a chatbot that responds to natural language inputs with empathetic text responses generated by pattern-matching scripts ([86]). GenAI refers to a set of artificial intelligence machine learning technologies that can generate content based on text prompts.

Recently, an unprecedented demand for AI technologies and skills has surged across various sectors, such as healthcare, finance, manufacturing, retail, and automotive. As more high-tech companies are investing significant resources to incorporate AI into their operations, the need for AI experts is expected to increase dramatically by 2025 ([4]). The employability report found that the workforce is unprepared, and 70 percent of graduates think generative AI should be incorporated into courses from the graduates’ perspective ([14]).

Within the application of GenAI in a higher education context, numerous studies explore universities’ reactions to GenAI usage in a higher education context. [39] ([39]) revealed that attitudes towards generative AI in top-ranked universities vary, including strong opposition or support for the responsible and ethical use of AI. [56] ([56]) found that with the adoption and rapid development of GenAI tools in higher education, Chinese universities have been using the technology to enhance teaching effectiveness, optimize research processes, improve student services, and more. In addition, the benefits of GenAI usage in higher education have been identified as adapting instruction to different types of students, providing feedback for students, developing course design materials for teachers, and supporting academic writing in several studies ([82]). The potential challenges of GenAI have also been found to be significant, such as bias and inclusion, as well as ethical and regulatory considerations ([69]).

Further research has been called for to gain a deeper understanding of the impact of GenAI use in HE classrooms on stakeholders’ responses, attitudes, and behaviors, to ensure successful integration between GenAI and higher education classrooms. Previous studies have examined the reactions, attitudes, and use of GenAI tools in general in the immediate aftermath of GenAI technologies. However, there is an urgent need for in-depth research on changes in students’ and teachers’ responses, attitudes, and behaviors based on authentic teaching and learning experiences, to provide an empirical base for identifying the fallout between technology and higher education integration to conduct research to bridge the gap.

This systematic literature review responds to this research gap, examining stakeholders’ responses, attitudes, and behaviors regarding GenAI applications. Notably, it delves into GenAI’s influence on teaching and learning in a classroom setting. The overarching questions for this study are as follows: what are students’ and teachers’ responses, attitudes, and behaviors about GenAI in HE classrooms? The following are three specific research questions:How does the use of GenAI for teaching and learning affect students’ and teachers’ responses in higher education classroom settings?How does the use of GenAI for teaching and learning affect students’ and teachers’ attitudes in higher education classroom settings?How does the use of GenAI for teaching and learning affect students’ and teachers’ behaviors in higher education classroom settings?

## 2. Materials and Methods

A systematic review methodology was used to answer the three guiding questions of this study ([65]). PRISMA was used to search, identify, and select articles for inclusion. Scholarly articles (e.g., conference proceedings, journal papers) published over the past 5 years were reviewed and analyzed.

The research began with the search for research articles to include in this study. Based on the research question, the study parameters were defined in terms of search years and the types of publications to be considered. Next, databases were selected. Boolean searches were created and used for databases. Once a set of publications was located from those searches, they were examined within GenAI’s inclusion and exclusion criteria. This was performed to determine which studies would be included in the final study. The relevant data to match the research questions were then extracted from the final set of studies and coded. This method section describes each method in full detail to ensure transparency.

### 2.1. Search Strategy

In this study, scholarly articles were selected according to a robust set of criteria. This ensured that the selected articles adhered to academic standards of rigor and relevance, as well as aligned with the study objectives. Specific keywords related to GenAI used in higher education teaching and learning were used to improve the reliability of the publication data generated. The final Boolean search string was as follows: 

TITLE-ABS ((“Generative AI” OR “generative artificial intelligence” OR “ChatGPT” OR LLM OR “GenAI”) AND (“pedagog*” OR “teach*” OR “learn*” OR course OR classroom OR curriculum) AND (“higher education” OR “universit*” OR HE OR undergraduate OR “graduate”)).

In order to locate the relevant articles, a systematic search was conducted on the following electronic databases: Scopus, Web of Science, and educational databases hosted by EBSCO. We selected these databases since they were considered the major publisher databases. Filters were limited to January 2020 to August 2024. Using previously used search terms, 1930 articles were found. All these articles were imported into Rayyan software for screening. After removing 616 duplicate records, we reviewed 1314 records’ titles and abstract according to inclusion and exclusion criteria, resulting in 137 articles. Excluding 7 articles that could not be found in full text, 130 full-text articles were then read and analyzed in accordance with inclusion and exclusion criteria. After screening the full text, 99 articles were left. Figure 1 shows the flow of a systematic literature search within the PRISMA framework.

### 2.2. Inclusion and Exclusion Criteria

The search criteria were designed to locate articles that focused on GenAI for teaching and learning in higher education classrooms. In terms of the research questions, a set of inclusion and exclusion criteria was adopted (see Table 1).

### 2.3. Screening

The screening process involved the following steps: (1) removing duplicate articles, (2) removing articles that did not meet the inclusion criteria based on the titles and abstracts, (3) reading the full texts and eliminating the articles that did not meet the inclusion criteria, and (4) extracting data from the final filtered articles (see Figure 1). The screening was conducted manually using the inclusion and exclusion criteria in Table 1. Finally, there were 99 articles remaining for inclusion in this systematic review.

To capture pertinent information from selected studies, we systematically extracted data. Based on [70] ([70])’s study, a structured form for data extraction was developed to facilitate the two-cycle coding process, which includes (1) attribute coding: logging information about the literature for the first cycle of analysis (e.g., countries, qualitative or quantitative research) and (2) pattern analysis: identifying similar data sets by grouping them and generating major themes, including responses, attitudes, behaviors, and relevance to the research questions. In order to extract and analyze the main information of the paper (based on the article data, descriptions of GenAI in classrooms, and attitudes and behaviors of students and teachers), a spreadsheet and MAXQDA were used.

### 2.4. Analysis

Qualitative content analysis was utilized for data examination ([62]). Two researchers were involved in the procedures, independently reviewing each article, achieving an inter-rater reliability of 92% (Cohen’s Kappa coefficient). Disagreements were resolved through discussion until agreement was reached. Table A1 shows all the code relevant to all the included articles (see Appendix A). 

## 3. Results

The first section of results presents the main characteristics of the included studies, such as country, year, publication type, and research method, which were summarized using descriptive statistics. The following sections present the findings according to each of the three research questions. The core keywords “response”, “attitude”, and “behavior” will be defined before the results of each section are discussed.

### 3.1. Summary of the Articles Included in the Review

Countries. The 99 studies took place across 42 countries on different continents of the world. These continents include Asia, Europe, and North America etc. This indicates a significant interest and contribution from Asian and European researchers in the field of higher education classrooms using GenAI tools.

Years. With the accelerated use of GenAI in higher education, it shows that the trend towards GenAI use in HE classrooms has greatly increased. The included articles focused on publications in 2023 and 2024, with 22 published in 2023 and 76 in 2024. In addition, one was published in 2022.

Documents by publication type. Among the included articles, 63.6% are journal articles, 33.3% are conference proceedings, and 3% are book chapters.

Research methods. It is revealed that the majority of the selected studies are mix-methods research articles (46.5%). Quantitative method research articles are also significantly represented (33.3%). Qualitative method research articles (20.2%) provide foundational cases and ideas for advancing understanding and practical applications.

### 3.2. RQ1: How Does the Use of GenAI for Teaching and Learning Affect Students’ and Teachers’ Responses in Higher Education Classroom Settings?

In this systematic review, we define “response” as the affective and behavioral reactions of subjects (e.g., teachers, students) to the use of GenAI in classroom situations in the context of higher education. Table 2 provides a snapshot of the current themes and priorities.

#### 3.2.1. Positive Responses

a) Positive Emotional Responses: The primary area of positive responses, with 73 studies, identified positive emotional responses of students and teachers to GenAI’s utilization. Participants usually mentioned these words, such as helpful, curious satisfied, confident, and enjoyable. For instance, [71] ([71]) showed that the ChatGPT assistant was highly rated by participants for its usability and helpfulness in assisting them with their learning process. [49] ([49]) incorporate ChatGPT and other large language models (LLMs) into a graduate-level Computational Biomedical Engineering course. Overall, students found the LLM component worthwhile because the material demonstrated that every student and instructor gained extensive experience with exposure to LLMs during this course. This increased their understanding of these tools’ capabilities and limitations. [89] ([89]) found that many teachers became quite enthusiastic about trying it out for themselves and with their classes after the chatbot’s integration into the course.

b) Willingness to Continue Using GenAI: A total of 32 out of 99 studies indicated that students and teachers might continue to use generative AI tools in the future or recommend them to others. For instance, [67] ([67]) introduced the AI Knowledge Assistant to address existing technology limitations by enhancing the quality and relevance of large language models (LLMs). In the feedback, students demonstrated a strong desire to employ the assistant in future projects. In [52] ([52])’s study, both groups of students had access to an LLM tutor interface to solve assignment problems. After solving the problems, one group was asked to engage in reflection through the same tutor interface. In the end, both groups were asked to rate how helpful they found the LLM tutor and whether they were willing to interact with the LLM again. Students who were given the opportunity to engage in reflection with the tutor reported a higher willingness to interact with the LLM Tutor, compared to students who only used the tutor to solve problems.

#### 3.2.2. Negative Responses

a) Negative Emotional Responses: A total of 46 out of 99 studies revealed that students and teachers had negative feelings towards GenAI tools, such as fear, frustration, lack of satisfaction, viewing them as unnecessary, distrust, etc. In [109] ([109])’s study, students reported negative experiences, considering ChatGPT interaction less satisfactory. [103] ([103]) found that many students’ experiences of satisfaction during the initial phase of the course changed as the course progressed. [1] ([1]) identified that some faculty members expressed concerns about the limitations of ChatGPT in handling edge cases and adapting to student needs.

b) Reluctance to Continue Using GenAI: Of the 99 studies, only 16 studies showed that students and teachers expressed hesitancy about potentially using GenAI in the future or recommending it to others. For example, 5 of 25 students who came from three courses aimed at providing pre-service teachers with practical training on teaching skills said that they might not continue to use GenAI to support teaching in the future ([58]). [89] ([89]) reported two kinds of negative reactions to GenAI that could limit the willingness of students to use it: (i) the fear that they would be punished and caught transgressing and (ii) a hesitancy to use it if it took away from the authentic expression of their ideas. [103] ([103]) illuminate how students’ engagement with AI impacts the resultant experience. Results indicate it is largely determined by students’ perceptions of AI’s effectiveness in fulfilling their learning needs. Curiously, in some cases, GenAI tools, while generally liked and seen as potentially improving the quality of assignments, were not very likely to be used for future tasks. For example, with regard to video-producing AI, while the respondents were neutral to satisfied with Fliki’s functionalities, it seemed unlikely that they would employ it for future multimedia projects. A respondent who was “not at all likely” to use the tool for enhancing their future presentations was also “moderately” satisfied with the functionalities of the text-to-video converter ([12]).

#### 3.2.3. Neutral Responses

Regarding GenAI being used in the classroom, 16 of 99 studies expressed neutral responses. In [71] ([71])’s study, regarding the helpfulness in solving coursework problems, there were also two neutral responses. [21] ([21]) revealed a near-even split in preference for GenAI-generated or human-created feedback, with clear advantages to both forms of feedback apparent from the data. [60] ([60]) discovered that participants who were more careful in their use of ChatGPT were likely to seek additional support and verify the accuracy of the generated code ([74]).

### 3.3. RQ2: How Does the Use of GenAI for Teaching and Learning Affect Students’ and Teachers’ Attitudes in Higher Education Classroom Settings?

In this systematic review study, we define attitude as a series of evaluations of GenAI tools, which are assumed to be derived from emotions, specific beliefs, and behavior intentions associated with those GenAI tools. The various attitudes toward GenAI used in classroom settings can be grouped into several distinct categories, as shown in Table 3.

#### 3.3.1. Positive Attitudes

a) Enabling students to adopt an entirely new and integrated approach to professional learning: The majority of studies (80 of 99) indicated that students and teachers reported advantages of GenAI utilization in HE classroom settings, including enhancing their understanding of the subject, deepening their understanding of specialized knowledge, and improving the learning efficacy of the subject matter. For example, [50] ([50]) suggested that the successful integration of ChatGPT as a teaching aid can provide novel insight into creative writing instruction. [81] ([81]) found that upper-division students use GenAI to generate and revise texts for chemistry writing, resulting in fewer grammatical errors, fewer extremely short and very long sentences, and improved readability. According to [5] ([5])’s study, GenAI tools are beneficial for students in science and engineering education to expand their creative horizons, improve their technical proficiency, and enhance the acquisition of knowledge. 

b) Reshaping students’ study methods and habits: This theme emphasizes the attitudes towards GenAI suggesting that it altered study habits and learning modes in numerous ways, as evidenced by the 55 studies related to this topic. As GenAI tools can provide immediate feedback, save time and effort in research and projects, facilitate collaborative learning, and support many other study approaches, the successful integration of GenAI as a teaching aid offers promising opportunities for providing personalized support for the reconstruction of learning habits and methods. For example, GenAI was found to have the capability to test students’ basic knowledge ([72]). As one student noted, AI tools offered a direct, focused approach which could alleviate their feeling of being overwhelmed by information. Some students noted that utilizing AI in research has greatly accelerated some traditionally lengthy parts of the process, which has allowed them more time for deeper analysis and critical thinking. ([6]).

c) Bridging classroom learning to real-world needs: In this review, 18 of 99 studies indicated that the integration of GenAI into university classrooms can bridge skill gaps between educational settings and workplace needs. This is particularly helpful in the context that employers often do not adopt effective training practices given concerns about the associated costs. GenAI tools can overcome these barriers by offering learning experiences for future professional development. For instance, [93] ([93]) indicated that ChatGPT can serve as a cost-effective learning and training tool for students and possibly for working professionals in the context of occupational safety.

d) Motivating teachers to update their teaching methods: Thirty studies suggest that GenAI tools like ChatGPT can serve as effective educational and training tools for teachers at universities to improve their teaching methods. In [1] ([1])’s study, the faculty questionnaire showed general faculty agreement that the AI assistant demonstrated strengths in certain areas, including accuracy of programming concepts, completeness of explanations, use of effective teaching strategies, and encouraging problem-solving. 

e) Encouraging teachers to maximize work efficiency: Eight studies revealed that incorporating GenAI into classroom settings can benefit the teaching process significantly, which not only reduces the burden on teachers, but also allows teachers to spend more time on more personalized teaching and guidance. For instance, according to [3] ([3]), the teaching staff felt that students often used ChatGPT in the class as a personal tutor, which allowed them to allocate their time more efficiently, focusing on more difficult questions.From the perspective of students, ChatGPT enabled them to ask teachers fewer questions directly, so teachers could focus on individuals with difficulties instead of wasting time on questions.

#### 3.3.2. Concerns

a) The low quality of generated content: Among the 99 studies, 44 studies were concerned about the low quality of the generated content. These concerns about GenAI included giving inaccurate information and providing vague advice. For instance, students using ChatGPT for their learning assignment confirmed that some information was outdated, significant content was missing, and part of the answer was occasionally incorrect ([45]). Another notable problem is the lack of depth, relevance, and creativity. For instance, participants pointed out that ChatGPT often generated responses failing to contain the needed details. As the participants felt that the innovation of the ideas provided by ChatGPT did not meet their expectations, they still needed to design more innovative situations to better fit the purposes of classroom teaching and learning ([58]). A further challenge was validating the source of the information ([30]). [45] ([45]) confirmed that links to references were nonexistent or not working.

b) The quality of generated content depends on various factors: Among the 99 studies, 34 studies focus on the fact that the quality of generated content is dependent on a variety of elements, including the precision and clarity of users’ prompts and the effectiveness of scholarly dialogues. [74] ([74]) indicated that GenAI results can vary depending on the prompts or wording used as input. This variability can lead to inconsistent or unexpected answers. [64] ([64]) target the weakness of ChatGPT’s overly generic text, indicating that it requires much more human prompting, cajoling, and manual editing to produce the desired results. [9] ([9]) indicated that when deployed in classrooms, or by students, users need to be educated about ChatGPT and similar GenAI tools on the following aspects: (1) The quality of the prompts drives the quality of the results. (2) Solutions and code snippets provided by ChatGPT are usually incorrect and need to be examined before use.

c) Ethical issues: Of the 99 studies, 23 studies addressed ethical issues related to GenAI-generated content, such as the authenticity of generated content, risk of plagiarism, and responsible use of generated content. With the aim of addressing ethical issues in pedagogical practice, it is recommended that content generated by GenAI in higher education classrooms be carefully integrated. In [6] ([6])’s study, one student expressed concern about potentially crossing ethical boundaries and compromising academic integrity, restraining him/her from exploring the use of GenAI tools. [81] ([81]) found that only one of the students who used ChatGPT formally acknowledged it in an Acknowledgments section, despite all being told that this was the formal standard at the start of the semester. Regarding the responsible use of GenAI, [88] ([88]) highlighted the potential benefits of AI for societal benefit, but these were sometimes tempered by concerns. One student expressed concerns about its influence on the degradation of human culture and the deepening of social inequality in the world.

d) Over-reliance: Another concern is the significant drawbacks of users’ over-reliance on GenAI, as illustrated in 34 studies. In [3] ([3])’s study, even when ChatGPT provided incorrect answers, a number of students accepted them without realizing they were incorrect, and failed to use traditional classroom methods to acquire knowledge, such as referencing books, tutorials, educational materials, or websites. Some studies have pointed out that when students lack basic knowledge of a subject and the ability to make comprehensive judgments, they will become over-dependent, leading to problems ([94]; [3]). [6] ([6]) found that students are apprehensive about the potential consequences of over-reliance on GenAI, fearing it could potentially distort or negatively impact the quality of their work and lead to reduced creativity and knowledge.

#### 3.3.3. Mixed Attitudes

Of the 99 studies, 30 studies reported mixed attitudes towards GenAI being used in HE classrooms. Firstly, while recognizing the advantages of GenAI, students and teachers also raised the unsatisfactory aspects of GenAI. For instance, in [88] ([88])’s study, teachers indicated the quality of the essays had improved compared with the previous years. When the essays were written without the aid of GenAI, they were more informative and context-providing, with fewer stylistic and grammar flaws, but the number of personal comments and observations decreased. Secondly, students and teachers felt that GenAI could be used as a teaching aid, but not as a substitute for the teacher’s role. For instance, [24] ([24]) created a chatbot interface and conducted a case study comprising a controlled experiment with 26 university software engineering students. One student noted that while the interactivity and response time provided by the chatbot were positive, the chatbot would never replace the educator. Finally, students and teachers responded that although they allowed GenAI to generate the content, they adjusted and improved it according to their own needs ([59]).

### 3.4. RQ3: How Does the Use of GenAI for Teaching and Learning Affect Students’ and Teachers’ Behaviors in Higher Education Classroom Settings?

In this systematic review study, we define behavior as the reactions to GenAI in terms of both observable activities and the underlying mental process that drives actions. The various learning and teaching behaviors of students and teachers are shown in Table 4.

#### 3.4.1. Promoting Student Learning Behaviors in Multiple Dimensions

a) Promoting learning behaviors in the cognitive processing dimension: Based on the 99 studies, 52 demonstrated that GenAI impacts learning behaviors in the cognitive processing dimension.

*Critical thinking.* GenAI may offer students feedback and guidance, which can help them develop a deeper understanding of the topic and enhance their critical thinking skills. For instance, a significant improvement in critical thinking was observed in the self-assessment of these skills following the intervention, which was the integration of a GenAI-based virtual assistant in a university-level network management course ([71]). For example, GCLA systematically offers hints rather than outright solutions, prompting students to refine their reasoning to reach the correct conclusion ([53]). One possible justification is that GenAI enhances critical thinking because they provide students with a more personalized learning experience, providing a different or broader perspective on the problem, adjusting the difficulty level and pace of instruction to meet students’ individual needs and learning styles ([2]).

*Creative thinking.* The potential of GenAI technologies in supporting student’s creativity has caught the attention of educators and students. The use of ChatGPT for university in-class activities is found to contribute to the development of students’ creative thinking abilities. ([22]). For instance, [17] ([17]) indicate that the introduction of ChatGPT not only enriches the teaching methods of creative writing but also significantly enhances students’ academic expression and innovative thinking. [53] ([53]) found that Guidance-based ChatGPT-assisted Learning Aid (GCLA) has the potential to boost creative outcomes, which is in agreement with research advocating for brainstorming and innovative thinking in higher education, unlike ChatGPT’s more deterministic approach. By creating a learning environment in which students are encouraged to think divergently, they can demonstrate creative solutions within creative ideas.

*Decision-making.* It has been observed that GenAI can provide instant feedback and personalized support to higher education learners, which enables them to make better decisions. As indicated by the students, ChatGPT provides comprehensive information quickly, which is helpful in assisting with the decision-making process. ([88]).

*Problem-solving skills.* GenAI has shown considerable promise in fostering problem-solving skills in the higher education context. For instance, high-achieving students demonstrated superior problem-solving abilities compared to their low-achieving peers, likely due to several factors ([16]). [71] ([71]) explore how a GenAI-based virtual assistant can be integrated into network management courses at universities. It was found that participants’ problem-solving skills improved statistically significantly following the intervention.

*Meta-cognition.* In addition to fostering the above learning behaviors, generative tools can also foster students’ meta-cognition. For instance, despite some technical and language challenges that surfaced, as well as skepticism from a small number of learners, post-training interviews indicated a shift towards improved metacognitive awareness and autonomous learning ([95]). The correlation between GenAI and classroom learning has been examined in some studies to explore potential avenues for metacognitive awareness. As soon as a problem is addressed, students are asked to reflect on their problem-solving process, formulate a critique in their own words, and report their individual observations and perceptions.

b) Enhancing learning behaviors in the self-management dimension: Thirty-one studies have shown that GenAI affects learning behavior in the self-management dimension.

*Self-efficacy.* Some studies show that students who engaged in the course using GenAI tools developed more positive learning behaviors, such as general self-efficacy and self-efficacy in specific areas. [58] ([58]) examined how general self-efficacy changes before and after GenAI use. After a four-week experiment, pre-service teachers in both the experimental group that utilized ChatGPT for assisted teaching and the control group that did not use ChatGPT for assistance both significantly improved their self-efficacy. Regarding specific self-efficacy, [36] ([36]) revealed that the experimental groups, particularly Group A (using the Companion System with a strategic tool for structuring and visualizing information), significantly outperformed the control group in boosting information literacy self-efficacy. The study introduced a novel approach to help participants increase their ethical self-efficacy and ethical decision-making, and the result demonstrated significant improvements in three crucial areas of ethical development: ethical self-efficacy, ethical decision-making confidence, and decision conflicts ([34]).

*Self-regulated learning behavior.* GenAI tools have emerged as a promising tool to reshape self-regulated learning experiences, poised to redefine autonomous learning. [90] ([90]) suggested that ChatGPT feedback quality is influenced by personal goals, learners’ self-regulation writing strategies, and multilingual capability. [104] ([104]) found that self-regulation levels were high among these three online ChatGPT groups. Furthermore, learning in peer and group environments developed self-regulation skills more effectively than learning independently. Those findings offer potential for exploring ways to leverage creative AI tools to facilitate students’ self-regulating learning strategies and behaviors.

*Learning motivation.* GenAI has been recognized as contributing to improved learning motivation. [53] ([53]) noted that traditional GenAI was slightly superior in terms of intrinsic and extrinsic motivation increases, particularly in higher education contexts. [98] ([98]) aimed to explore the impact of GPT feedback in a VR learning environment. The results showed that, compared to the control group receiving traditional feedback, the experimental group using GPT feedback exhibited significantly greater improvements in learning motivation. However, the results were not consistent with extrinsic motivation. [55] ([55]) analyzed the extrinsic motivation ratings of the two groups, and the results did not indicate a significant difference.

c) Promoting learning behaviors in the social interaction dimension: According to 39 of the 99 studies, GenAI impacts learning behaviors in the social interaction dimension.

*Engagement.* Active engagement practices are seen as a valuable tool to support the “learning-by-doing” pedagogy, which is vital in higher learning environments ([53]). This was evidenced by the students’ positive reception and the notable improvement in learning retention and engagement ([78]). [53] ([53]) introduced the GCLA, which has facilitated more effective learning experiences in blended learning environments. It has also ensured that students take an active part in their educational journey.

*Collaborative skills.* Collaborative skills to maintain a relationship with others refer to the social aspects of learning, which are considered to be necessary for a learning environment. [78] ([78]) suggested that participants experienced increased engagement, better teamwork dynamics, and enhanced collaboration. Based on the findings, students expressed optimism about ChatGPT’s effectiveness in enhancing classroom engagement, fostering collaboration, and fostering active participation ([95]).

#### 3.4.2. Facilitating Teachers to Optimize Teaching Behaviors

a) Guiding teachers in innovating teaching methods: Among the 99 studies, 20 studies demonstrated that GenAI complements teachers’ teaching behaviors in several ways. These studies aim to examine GenAI’s effects on supporting students’ or pre-service teachers’ pedagogy, including their ability to design and prepare classes. For instance, [58] ([58]) conducted a study aiming to offer practical training in teaching skills, encompassing lesson preparation, classroom instruction, and post-class analysis to third-year pre-service teachers. The experimental group utilized ChatGPT to assist with teaching, whereas the control group did not employ ChatGPT for support. [78] ([78]) conducted an 8-week quasi-randomized study. The experimental group members participated in small-group collaborative learning with ChatGPT assistance, while the control group members engaged in traditional collaborative learning. A total of 36 sophomore pre-service teachers were assigned to create teaching aids and conduct simulated instructional sessions. 

b) Supporting teachers in enhancing assessment practices: Given the prominence of GenAI generators like ChatGPT, which are noted for their simplicity and accessibility, out of the 99 studies, 10 studies examined how GenAI tools complement teacher assessment, and the overall effectiveness of this combined approach. For instance, by shedding light on the synergy between ChatGPT and teacher assessments, [59] ([59]) uncovered three ways in which ChatGPT complements teacher assessment, benefiting students at various writing proficiency levels: (1) fostering deeper comprehension of teacher assessments among students, (2) encouraging students to make judgments regarding feedback, and (3) promoting independent thinking about revisions.

#### 3.4.3. The Task Behavior Performance Is Unstable

a) Generative AI promotes the task behavior performance: Among the 99 studies, 59 studies found that participants in groups that used GenAI performed better in the experiments or tasks studied than those in groups that did not use GenAI or those using traditional pedagogical methods. These studies reveal that the use of GenAI in the classroom positively affects students’ or teachers’ learning and teaching behaviors.

b) Generative AI does not promote task behavior performance: Among the 99 studies, 25 studies found inconsistent findings compared with previous studies. For instance, [7] ([7]) conducted pre- and post-tests to assess changes in digital storytelling skills and cognitive load, which revealed no significant improvement in digital storytelling skills for the experimental group compared to the control group. [50] ([50]) showed that students’ performance was not influenced by ChatGPT usage (no statistical significance between groups), nor were the grading results of practical assignments and midterm exams affected. [43] ([43]) suggested that students who used LLMs more frequently for code generation received a lower final grade. One study found that students trusted human teachers more than computers and preferred dialogue with them ([107]).

## 4. Discussions

From initial entries in the selected database, 99 studies met the inclusion criteria and were investigated in this systematic review. We aimed to determine the current state of research on students’ and teachers’ responses, attitudes, and behaviors toward GenAI usage in higher education. 

As a result of our study, this review revealed a range of study responses of students and teachers towards the use of GenAI in HE classroom teaching and learning (see Table 2). These responses can be classified into three main categories: positive responses, negative responses, and neutral responses. The majority of studies (73 out of 99) in this review overwhelmingly reflected positive emotional responses (e.g., enthusiastic, excited, enjoyable) to GenAI usage in HE classroom settings. It is noteworthy that students and teachers in some studies, although their emotions were on the positive side, indicated that their willingness to use GenAI was low or even that they had no intention of continuing to use GenAI in the future. The reason for this seemingly inconsistent result may be due to the fact that some of these studies did not involve an investigation of willingness to use. However, willingness to use is a prerequisite for actually using GenAI and discovering its shortcomings and potential. This warrants further exploration of the factors that influence it in the future.

Three types of attitudes about GenAI used for learning and teaching practices emerged from certain studies (see Table 3), including positive attitudes, concerns, and mixed attitudes. From a positive perspective, GenAI can benefit students’ professional learning and study habits. But this systematic review found that relatively few studies have explored teachers’ attitudes toward GenAI compared to considering students’ attitudes. How can teachers leverage the benefits of GenAI in higher education classrooms? How can teachers better connect classroom learning to real-world needs? Responding to these questions requires more attention to the integration of GenAI into teaching methods and educational assessment in future research. Several concerns have been raised about GenAI usage in HE classroom settings. Studies concerned with the quality of generated content and its influencing factors, especially prompts, dominate. But as the problem of over-reliance is taken seriously, the number of studies in this systematic review study that raise concerns about ethical issues is still relatively small, and much smaller than the number of studies expressing positive attitudes toward GenAI. Therefore, in a context where generative technologies are still evolving, universities urgently need to invest in research and formulate policies surrounding ethical issues. Finally, mixed attitudes towards GenAI usage in HE classroom settings appear to be rational. For example, they agree that GenAI has advantages that traditional teaching methods and tools do not have, but also, they suggest that learners and educators need to weigh both the advantages and disadvantages of GenAI and improve the use of the generated content according to real learning and teaching needs.

Due to its significant features of interaction and dialogue, GenAI can also act as a “pedagogical catalyst” to support and empower the behaviors of students and teachers (see Table 4). In addition to promoting students’ learning behavior in cognitive processing, self-management, and social interaction dimensions, GenAI usage is also evidenced to have a positive impact on innovating teachers’ teaching methods and assessment practices. However, the use of GenAI in HE classroom settings does not always result in positive teaching and learning outcomes. Taking critical thinking-oriented behavior as an example, in some studies, students’ critical thinking-oriented behavior was significantly increased by GenAI usage, while in others, students reported that GenAI instead hindered their critical thinking growth. One possible illustration of the significance of GenAI in enhancing critical thinking-oriented behaviors is that it offers students the possibility to engage in dialogue with GenAI, fostering a discussion-based learning environment that enables students to develop robust critical thinking skills. However, when users lack basic subject matter knowledge and the ability to judge, they can rely uncritically and excessively on generated content, which hinders their creative behaviors and may even cause them to absorb misleading information.

Overall, this review contributes to the growing body of knowledge about critical areas of focus on the impact of GenAI potential use on students’ responses, attitudes, and behaviors across various disciplines. In practice, the results benefit stakeholders and the research community by enabling the use of current GenAI tools for learning and teaching.

## 5. Conclusions

This systematic literature review is a valid foundation for future studies. It is imperative to advocate for a robust body of empirical research examining the multifaceted impact of GenAI on students’ and teachers’ responses, attitudes, and behaviors from an interdisciplinary and multidimensional standpoint.

It is worth stating that this paper reviews the literature in the English language, meaning significant research in other languages was left out. This limitation means our study might have missed non-English GenAI topic trends. To enhance inclusivity, it is crucial to expand representation across journals, particularly non-English publications. Additionally, restricting the search term to “ChatGPT” could have inherent biases, limiting the inclusion of more literature related to other generative AI tools. Future research should aim to include more diverse GenAI tools and explore the impacts of their application in higher education teaching and learning to enhance the robustness and reliability of the findings. 

A common theme among these studies is integrating generative AI into curriculum and instruction, emphasizing the role of GenAI in facilitating students’ learning through deeply personalized learning experiences, as well as exploring GenAI’s potential to assist teachers with course design, assessment of learning, and other pedagogical behaviors. The findings suggest several avenues for future research.

As can be seen from the results, GenAI serves as “scaffolding” to guide, support, and empower students and teachers’ learning activities and teaching processes. As education is about serving life and living, future research should focus on the autonomy and creativity of students and teachers when interacting with GenAI tools.Most of the included studies mentioned responses, attitudes, and behaviors about GenAI from the perspective of students (as can be seen in Table 2, Table 3 and Table 4). For future research, it would be worthwhile to further study this topic from a teacher’s perspective. For instance, what factors influence teachers’ willingness to use GenAI for innovating their teaching method? How do teachers develop syllabi for GenAI courses? And how do teachers increase their own GenAI literacy?It is revealed that many of the selected studies (46.5%) are mixed-methods research articles. Most of them are based on quasi-experimental research, questionnaires, interviews, textual analysis, and other methods. It is imperative that more interdisciplinary studies are conducted in the future, including integrating psychological and neuroscientific research methods, to detect changes in the psychological state, cognitive processing process, and brain activation state of individuals during generative AI–human interactive teaching in real time and simultaneously.Future research should pay more attention to students’ and teachers’ negative responses and concerns about GenAI, which are less explored compared to positive responses and attitudes in this systematic review (as can be seen in Table 2 and Table 3). In order to produce positive outcomes for students and educators in the age of artificial intelligence, it is considered that universities should take the initiative to facilitate the rational and ethical use of GenAI tools by emphasizing the humanistic and emotional dimensions of GenAI, not only to leverage the “advantage” of GenAI but also to promote the “goodness” of the technology.

Our systematic review reflects the responses, attitudes, and behaviors of students and teachers about GenAI for learning and teaching in HE classrooms. It also underscores the critical need for further research to overcome GenAI integration barriers. By addressing these issues and leveraging the full potential of GenAI, universities can reshape learning modes and teaching philosophies, and push the boundaries of disciplinary knowledge.

## Figures and Tables

**Figure 1 behavsci-15-00467-f001:**
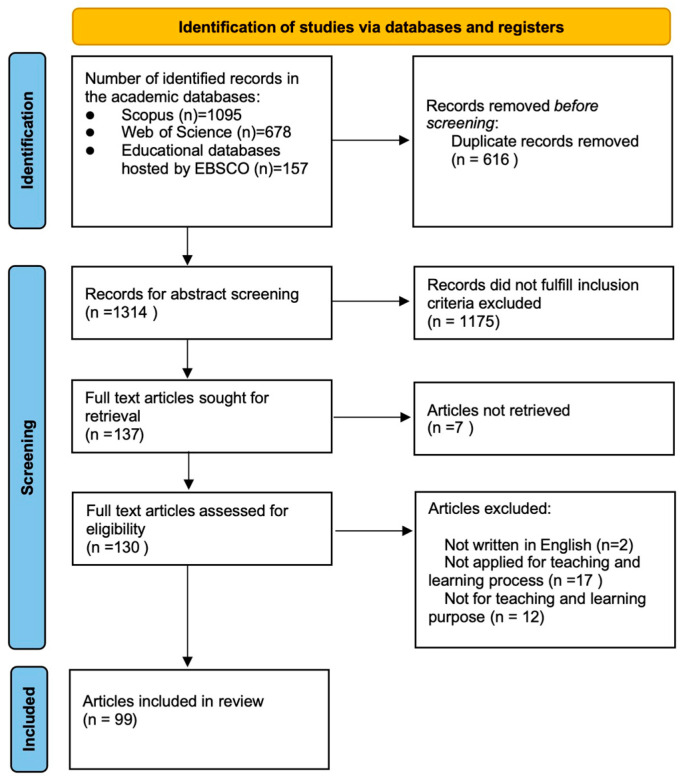
RISMA flow diagram of the systematic review.

**Table 1 behavsci-15-00467-t001:** Inclusion and exclusion criteria.

Inclusion	Exclusion
Published 2020–August 2024	Published before 2020
Written in English	Not written in English
Published in journals, conference papers, book chapters	Book, abstract, thesis, PhD dissertation
Empirical research, case study	Theoretical or conceptual papers, review articles
Has thematic focus on GenAI in HE	No thematic focus on GenAI in HE
Use of GenAI in classroom setting Research for teaching and learning purposes	No use of GenAI in classroom setting Not research for teaching and learning purposes

**Table 2 behavsci-15-00467-t002:** Responses with GenAI used for learning and teaching practice.

Theme	Sub-Theme	Examples	Number of Studies
Positive responses	Positive emotional responses	Another student commented: “I enjoyed working with ChatGPT, because I got to learn and understand something that is going to be a part of the future.” ([49])	73
	Willingness to continue using GenAI	When asked “Will you continue to use ChatGPT and other generative AI for assisted teaching in the future?”, a total of 20 of the 25 interviewees said that they would continue to use ChatGPT and other generative AI, while 5 indicated that they might not use them to support teaching in the future. ([58])	32
Negative responses	Negative emotional responses	“I wasn’t interested in AI and didn’t feel the need to get to know it. (R5, F).” ([88])	46
	Reluctance to continue using GenAI	Notably, of the 12 total responses, only 3 expressed hesitancy about potentially using ChatGPT to study English in the future. ([95])	16
Neutral responses	--	Cautious: This student is more careful in their use of the ChatGPT and is less confident in the generated code. They are likely to seek additional support and verify the accuracy of the code produced. ([60])	16

**Table 3 behavsci-15-00467-t003:** Attitudes with GenAI used for learning and teaching practice.

Theme	Sub-Theme	Example	Number of Studies
Positive attitudes	Enabling students to adopt an entirely new and integrated approach to professional learning	Typical TSs in agreement with the benefits arising from the inclusion of ChatGPT in the educational context include “Being highly optimistic about ChatGPT and this activity of creating short stories, I believe it can be utilized to compose short stories based on historical events or scientific facts, which can be incredibly engaging for interdisciplinary work in the classroom” (Prof_13f) and “I thought ChatGPT was highly effective in creating meaningful texts” (Prof_2f). ([25])	80
	Reshaping students’ study methods and habits	Some students identified potential benefits of technologies such as ChatGPT. In written comments, some likened it to a tutor or a professor who was constantly available and who could quickly explain concepts in new and helpful ways. ([100])	55
	Bridging classroom learning to real-world needs	“I really appreciated the ChatGPT activity because it allowed me to practice applying sampling techniques to real-life scenarios. It helped me feel more confident in my understanding of the material”. (participant 4) ([22])	18
	Motivating teachers to update their teaching methods	The Business Game category summarizes students’ evaluations and reflections on the effectiveness of the simulated business game as a learning method. This category includes insights into the dynamics of the game, the role of interaction with ChatGPT, and the overall evaluation of the simulation as a pedagogical tool. Students described the simulation as an innovative and interactive way to understand complex cloud migration concepts and processes. ([85])	30
	Encouraging teachers to maximize work efficiency	The most frequently mentioned functions of GenAI tools were generating teaching and learning materials, checking assignments, correcting grammar mistakes, and designing lesson plans. They maintained that these functions of GenAI tools could help them “reduce the workload”, “increase efficiency”, “brainstorm ideas”, and “refresh thoughts”. ([66])	8
Concerns	The low quality of generated content	Students felt that the knowledge of the Chat Generative Pre-Trained Transformer is limited and not very reliable. Images and videos were not available on the given topic on ChatGPT; hence, students used Google and other internet applications to search for answers. ([11])	44
	The quality of generated content depends on various factors	An interview extract exemplified the participants’ experiences. “ChatGPT pointed out that the scope of my prompt was too broad. In response, I rephrased the prompt and focused on one aspect at a time. As a result, ChatGPT provided me with relevant information and guidance, which helped me form ideas and arguments effectively”. The participants noticed that increased interaction with ChatGPT led to high-quality feedback. ([90])	34
	Ethical issues	For RQ2 (What are the pedagogical and technological outcomes of using generative AI to reflect on the nursing profession?), the themes identified included the following: (1) self-reflection on students’ moral and professional identity development, (2) strange and inaccurate images, and (3) biases/stereotypes of nurses not based on contemporary realities. ([80])	23
	Over-reliance	Concerns included AI’s impact on privacy, ethics, developer education, and reliability. Over-reliance on AI possibly hindering skill development (P3) and AI code’s reliability (P7) were other concerns. ([99])	34
Mixed attitudes	--	Thus, as a course designer (echoing Yaron’s experience above), Yasemin does not think ChatGPT is likely to render educators or education designers obsolete. ChatGPT is a convenient tool for educators requiring unit outline generation; however, without the input of an experienced educator or education designer, it does not currently appear to have the capacity to create a learning unit on its own. ([64])	30

**Table 4 behavsci-15-00467-t004:** Behaviors with GenAI used for learning and teaching practice.

Theme	Sub-Theme	Examples	Number of Studies
Promoting student learning behaviors in multiple dimensions	Promoting learning behaviors in the cognitive processing dimension (e.g., meta-cognition, critical thinking, creative thinking, decision-making, problem-solving)	Students in the experiment group (EG) used ChatGPT for in-class tasks, while students in the control group (CG) used traditional databases and search engines. Compared to the CG, the EG demonstrated a significant increase in critical thinking, reflective skepticism, and critical openness compared to the CG. ([22])	52
	Promoting learning behaviors in the self-management dimension (e.g., self-regulation, self-efficacy, motivation)	Regarding the utilization of ChatGPT, students reported a rise in their usage of ChatGPT to boost their self-efficacy in critical thinking (from 46% to 67%, *p* = 0.31) and would recommend using ChatGPT as a tool for others to enhance their critical thinking skills (from 72% to 88%, *p* = 0.39). ([30])	31
	Promoting learning behaviors in the social interaction dimension (e.g., engagement, collaborative skills)	In addition, peer collaboration became evident as students enjoyed discussing and exploring the functionalities of ChatGPT together. “This time made me think about the best way to practice English. Usually I just followed what my teacher said, but now I can talk to my friends and make my own plan”. ([94])	39
Enabling teachers to optimize teaching behaviors	Guiding teachers in innovating their teaching methods	Specifically, participants showed a deeper understanding and more detailed elaboration when using GenAI tools to facilitate the planning process. There was a salient change from considering GenAI as a direct lesson plan generator to using it to refine existing plans and create classroom activities. The shift might be attributed to the instructor’s explicit guidance on using GenAI to refine lesson plans, coupled with targeted assignment tasks that encouraged such an application. ([66])	20
	Supporting teachers in enhancing their assessment practices	After using ChatGPT-4 for preliminary marking, the average marking time for each poetry assignment was reduced from 30 min to 10 min. This change greatly improved teaching efficiency. ([57])	10
Inconsistent task behavior performance	Generative AI promotes the task behavior performance	In the quantitative analysis part, we saw a skew toward higher points for students who were actively using the part of the LMS with AI-generated content. ([72])	59
	Generative AI does not significantly promote the task behavior performance	Table 4 shows the results of both groups’ performance in lab work. The average lab work success of Group I, which used ChatGPT, was 65.27%, whilst the average score of Group II (no ChatGPT) was only slightly better, at 66.72%. ([50])	25

## Data Availability

Data are contained within the article.

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
