# Peer review of "A Systematic Review of Responses, Attitudes, and Utilization Behaviors on Generative AI for Teaching and Learning in Higher Education"

_behavsci, 2025, doi:10.3390/bs15040467_

Round 1
Reviewer 1 Report
Comments and Suggestions for Authors
The paper has inherent bias for ChatGPT. It should have been searched and concluded rather than assumed that its the only one to be considered for the study?
The comparisons and results should also be presented in tabular form as well, that will help reader to comprehend better.
The basis of conclusion is not clear. There needs to be based on statistical comparison (e.g. sentiment/thematic analysis or any other approach ) for the literature being studied to objectively conclude the observations.
Comments on the Quality of English LanguageThere are some occassional grammar correction needed.
Some of the sentences can be made clearer. e.g. in the sentence "This growth has outpaced the availability of talent, leading to a significant gap." , "This growth" is vague without clear antecedent context.
A through review and/or english grammar checking tools could also be helpful.
Reviewer 2 Report
Comments and Suggestions for Authors
The flow of ideas in the Introduction was difficult to follow. Perhaps the authors could begin with the definition of GenAI – the popularity of GenAI resulting in a significant increase in investment in AI resources – research in the application of GenAI in higher education. There were sentences in the first paragraph of the Introduction which detracted from the flow of the argument.
2.5 should be placed in the Results section instead.
How were the definitions for responses, attitudes and behavior used in interpreting the research? If there is no specific function for these lengthy definitions, it might suffice to just provide one definition for each of the constructs instead.
Line 97 – Do present the codes in table format.
Line 208 - Please provide further elaboration on King et al (2024) – why was the LLM component worthwhile?
Line 218 – Why was only one group asked to reflect on the tutor interface? Further elaboration on the treatment options for the study for each group of students could be provided.
Line 236 – Was the issue with students being undergraduates/postgraduates or was the issue with the ability to prompt? This was unclear.
Line 242 – Why were they unlikely to employ Fliki for future multimedia projects?
The authors mentioned that the review examined students’ and teacher’s responses, attitudes and behaviors. However, the results section mainly presented the students’ perspectives. Little mention was made regarding the teachers’ perspectives for RQ1.
Line 395 – Do you mean the loss of critical thinking skills instead? Or the lack of independent learning?
Line 441 was unclear – what changed exactly?
Discussion – This section discusses the findings for RQs 1, 2 and 3 taken together. Please address contrasting findings across RQs 1,2 and 3. For example, some findings showed that GenAI fostered critical thinking while others revealed that it did not – explain factors contributing to these differences in this section. The contributions of this paper need to be clear in this section.
Comments on the Quality of English Language
There were many instances where the sentences were difficult to understand. Perhaps the paper could be edited for language. The paper could also be written in a more concise manner.
Reviewer 3 Report
Comments and Suggestions for Authors
Please see attached file

Round 2
Reviewer 2 Report
Comments and Suggestions for Authors
The authors have revised the manuscript based on the feedback provided. The manuscript is clearer now.
Perhaps the content in the table could be formatted flushed left instead of centralized.
Comments on the Quality of English LanguageThere were language issues in the manuscript, but this did not affect clarity.
